# Deep Characterization of a Greek Patient with Desmin-Related Myofibrillar Myopathy and Cardiomyopathy

**DOI:** 10.3390/ijms241311181

**Published:** 2023-07-06

**Authors:** Constantinos Papadopoulos, Edoardo Malfatti, Corinne Métay, Boris Keren, Elodie Lejeune, Julien Buratti, Sophia Xirou, Margarita Chrysanthou-Piterou, George K. Papadimas

**Affiliations:** 1First Department of Neurology, Eginition Hospital, Medical School, National and Kapodistrian University of Athens, ERN, EURO NMD, 11528 Athens, Greece; constantinospapadopoulos@yahoo.com (C.P.); sopxir@hotmail.com (S.X.); margarita.chrysanthou@gmail.com (M.C.-P.); 2Centre de Référence de Pathologie Neuromusculaire Nord-Est-Ile-de-France, Université Paris Est, U955 INSERM, EnvA, EFS, IMRB, F-94010 and APHP, Henri Mondor Hospital, 94010 Créteil, France; edoardo.malfatti@aphp.fr; 3APHP, Unité Fonctionnelle de Cardiogénétique et Myogénétique Moléculaire et Cellulaire, Centre de Génétique Moléculaire et Chromosomique, INSERM, Institut de Myologie, Groupe Hospitalier La Pitié-Salpêtrière-Charles Foix, Sorbonne Université, 75013 Paris, France; corinne.metay@aphp.fr; 4APHP, Centre de Génétique Moléculaire et Chromosomique, UF Génétique du Développement, GH Pitié-Salpêtrière, 75013 Paris, France; boris.keren@aphp.fr (B.K.); elodie.lejeune@aphp.fr (E.L.); julien.buratti@aphp.fr (J.B.)

**Keywords:** desmin, myofibrillar myopathy, cardiomyopathy, ribonucleic acid sequencing

## Abstract

Desmin is a class III intermediate filament protein highly expressed in cardiac, smooth and striated muscle. Autosomal dominant or recessive mutations in the desmin gene (*DES*) result in a variety of diseases, including cardiomyopathies and myofibrillar myopathy, collectively called desminopathies. Here we describe the clinical, histological and radiological features of a Greek patient with a myofibrillar myopathy and cardiomyopathy linked to the c.734A>G,p.(Glu245Gly) heterozygous variant in the *DES* gene. Moreover, through ribonucleic acid sequencing analysis in skeletal muscle we show that this variant provokes a defect in exon 3 splicing and thus should be considered clearly pathogenic.

## 1. Introduction

Desmin is a class III intermediate filament highly expressed in cardiac, smooth and striated muscle. This 53 kDa protein has a central a-helical coiled-coil rod domain, consisting of two seriatim a-helical segments, coil 1 and coil 2, that are connected by a ‘‘linker’’, and is flanked by a non-a-helical amino-terminal domain (so-called “head”) and a carboxy-terminal domain (so-called “tail”). The non-a-helical amino-terminal domain is essential for protein formation and the carboxy-terminal is important for protein-protein interplay. In striated muscle, desmin is located at the Z-disks, at the myotendinous and neuromuscular junctions and at the subsarcolemmal space, while in the heart muscle it is located at intercalated disks of the cardiomyocytes and Purkinje fibers. Desmin links the contractile apparatus to the sarcolemma and various cytoplasmic organelles, such as the nucleus, mitochondria, and lysosomes and promotes communication among the extracellular matrix and cellular organelles. Moreover, it maintains structural integrity and provides muscle cell resistance to externally applied mechanical forces [1,2,3].

Autosomal dominant or recessive mutations in the desmin gene (*DES*) result in a variety of diseases, including cardiomyopathies and myofibrillar myopathy, collectively called desminopathies [4,5]. Desminopathies are the best-studied disease entity within the clinically and genetically heterogeneous group of myofibrillar myopathies (MFM) [6]. The incidence and prevalence of desminopathies are currently not available. However, desminopathies are considered rare diseases, thus affecting no more than five individuals in 10,000 [1]. Within a group of 53 patients from 35 Spanish MFM families, desmin mutations were the second most frequently encountered gene defect [7]. Skeletal myopathy in desminopathies usually presents by the age of 30 years, but cases with early-or later onset have been reported. Usually, there is distal lower limb weakness and atrophy, involving the anterior compartment that later spreads to involve proximal and axial musculature (paraspinal or neck flexor muscles), but limb-girdle and scapuloperoneal presentations can also occur [4,8,9]. Cardiac involvement in desminopathies is predominant and variable as it may include arrythmias, hypertrophic, dilated, restrictive and non-compaction cardiomyopathy that may precede, occur at the same time or follow the appearance of clinical evident skeletal myopathy [10,11,12,13,14,15,16]. Desmin mutations account for up to 2% of pure dilated cardiomyopathies [17]. There is no clear correlation between the skeletal muscle and heart involvement in desminopathies. But the latter is associated with high morbidity and mortality and warrants close monitoring, even in asymptomatic individuals [18]. Respiratory insufficiency is not unusual, leading sometimes to the use of non-invasive ventilatory support, while some mutations have been correlated with severe and predominant respiratory muscle weakness [19,20]. Muscle imaging in desminopathies reveals early fatty infiltration of lower leg muscles, initially the *peroneus lateralis* and subsequently the *tibialis anterior* and the posterior compartment muscles. In the thighs, there is involvement of the *sartorius*, *gracilis* and *semitendinosus* muscles and relative sparing of the *semimembranosus*, even at advanced stages of the disease [19,21]. Characteristic light microscopy findings in skeletal muscle biopsies from patients with desminopathies encompass the presence of protein aggregates of rimmed and non-rimmed vacuoles, and areas of irregular internal architecture in muscle fibers (rubbed-out fibers and core-like areas), in conjunction with mild to severe myopathic features. Electron microscopy has a central diagnostic role in documenting signs of myofibrillar degeneration, as well as protein aggregation [1,16]. The typical ultrastructural findings include the presence of granulofilamentous accumulations, Z-disk streaming and sandwich-like formations of granulofilamentous material facing the Z-disks and mitochondria alongside it [22].

Pathogenic mutations are spread all over the entire *DES* gene and tend to cluster in exon 6, encoding the C-terminal of the coil 2 domain. Regarding the genotype-phenotype correlations, it seems that variants affecting the coil domain are usually associated with a predominant muscle phenotype, while if they affect the tail and head domains there is a predominant heart involvement. Most frequent *DES* mutations are missense mutations, causing single amino-acid substitutions, while small in-frame deletions and frame-shift mutations have only been reported in case reports [1,8,23]. Dalakas et al. first reported a splice site mutation (c.735+3A>G) responsible for the deletion of exon 3 in a patient with myopathy and cardiomyopathy [24]. Subsequently, a number of splice site mutations in the *DES* gene have been reported, most of them residing in the highly conserved among species, acceptor and donor splice site of exon 3 [18,25,26,27].

Here we describe the clinical, histological and radiological features of a Greek patient with myofibrillar myopathy and cardiomyopathy linked to the c.734A>G,p.(Glu245Gly) heterozygous variant in the *DES* gene. Moreover, we provide evidence for the pathogenicity of this variant through ribonucleic acid sequencing (RNAseq) analysis in skeletal muscle, and we show that this mutation is pathogenic by causing a defect in exon 3 splicing.

## 2. Case Report

The index case is a 36-year-old man born at term, following an uneventful pregnancy and a normal delivery, to non-consanguineous healthy parents. His psychomotor development was normal and he reported being athletic as a child. At the age of 18 years, during his military service, the patient was diagnosed with restrictive hypertrophic cardiomyopathy and a year later a pacemaker was implanted due to syncopal episodes. At the age of 23 years, he was diagnosed with atrial fibrillation complicated by an ischemic stroke with no residual neurologic deficit. Since the age of 28 years, he has reported difficulties climbing stairs and getting up from a squat. He was firstly seen at the neuromuscular clinic at the age of 33 years, and he presented bilateral *scapula alata*, and a waddling and stepping gait. There was mild weakness in the upper limbs (*biceps* and finger extensor muscles graded 4/5 on the MRC scale) and more severe in the lower limbs (hip extensors and *psoas* muscles 1/5 on the MRC scale, hip abductors, *biceps femoris* 3/5 on the MRC scale). Distally, there was involvement of *tibialis anterior* and *peronei* muscles (3/5 on the MRC scale). His serum creatine kinase levels were mildly elevated at 400 UI/L (normal values < 180 U/L). Needle electromyography demonstrated small amplitude, short duration, polyphasic motor unit action potentials, with early recruitment and small amounts of fibrillation potentials and positive sharp waves, as well as myotonic discharges in all examined muscles. A lower limb muscle computed tomography revealed fatty degeneration of *semitendinosus*, *gracilis*, *sartorius* and *adductor longus* muscles in the thigh and diffuse fatty degeneration of leg muscles (Figure 1). 

A left *vastus lateralis* muscle biopsy revealed increased muscle fiber size/diameter variation, with the presence of grouped atrophic fibers giving a pseudo-neurogenic aspect. There were prominent and multiple nuclear internalizations, and with both H&E and modified Gömöri trichrome staining we observed the presence of multiple rimmed vacuoles in about 1–2% of muscle fibers and rare cytoplasmic bodies (not shown). Histoenzymatic oxidative reactions were remarkable for the presence of irregular intermyofibrillar network irregularities giving a moth-eaten aspect to the fibers, and the presence of variably located areas, sometimes central, devoid of oxidative activity, in 10% of muscle fibers, as well as rubbed-out areas with less delimited borders. Immunohistochemistry for a battery of antibodies against protein involved in myofibrillar myopathies revealed the presence of desmin and myotilin (not shown) sarcoplasmic immunoreactive aggregates. An ultrastructural study showed the presence of subsarcolemmal, sarcoplasmic, granulofilamentous material (Figure 2) and Z-line streaming (not shown). 

Targeted sequencing of a multigene panel for myofibrillar myopathies revealed the missense mutation c.734A>G,p.(Glu245Gly) in the *DES* gene (NM_001927.4) in the heterozygous state. This missense variant localized before the last nucleotide of exon 3 in the coiled 1B domain of the protein has already been detected in an Indian family with a dominant inheritance but with an unknown effect [28]. Its allelic frequency is not reported in GnomAD (v1.3). The LOVD database reports it as of uncertain significance. Bioinformatics prediction tools predicted a splice effect. We performed RNAseq on an RNA sample from a quadriceps muscle biopsy. A defect in exon 3 splicing was observed in the heterozygous state with 3489 reads of the exon 2 to exon 4 junction, compared to 3829 reads for junction 2–3 and 3964 for junction 3–4. In comparison, we did not observe the 2–4 junction among controls consisting of RNAs from another muscle biopsy used in the same run (Figure 3). 

The precise deletion breakpoints determined by BAM data from RNAseq showed the deletion of exon 3 in the reading phase (r.640_735, hg38) in IGV software (Integrative Genomic Browser, version 2.11.1).

## 3. Discussion

Here, we describe a patient with restrictive hypertrophic cardiomyopathy, cardiac conduction defects and predominately limb-girdle muscle weakness carrying the c.734A>G,p.(Glu245Gly) heterozygous variant in the *DES* gene. Moreover, we provide evidence on the consequences of this variant at the protein level through mRNA sequencing on the skeletal muscle.

The propositus presented with restrictive hypertrophic cardiomyopathy and cardiac conduction defects, leading to a pacemaker implantation followed a few years later by the development of skeletal myopathy. Desminopathies are among muscle disease with prominent cardiac involvement [29] and should always be suspected in patients presenting in the third or fourth decade of life with myopathy and evidence of heart pathology [16]. Heart disease in desminopathies may precede muscle weakness by years, has variable presentations, comprising conduction system defects, arrhythmias and all forms of cardiomyopathy and can be severe enough to lead to heart transplantation, or, as in our case, to the implantation of a pacemaker. Cardiac involvement is not, clearly, correlated to the type of *DES* mutation nor with the severity of muscle disease and is the major determinant of the disease prognosis. All patients, as well as at-risk asymptomatic carriers of *DES* mutations, should be offered in-depth and regular cardiac investigations that should include a 12-lead surface ECG, a 24-h Holter ECG and a transthoracic echocardiography [1,4,18,30] and probably a heart MRI that has been shown to be more sensitive in detecting cardiac muscle involvement in early, asymptomatic stages [31]. 

In the framework for his muscle disease, the patient underwent lower limb computed tomography showing predominant involvement of *semitendinosus*, *gracilis*, *sartorius* and *adductor longus* muscles in the thigh and diffuse fatty degeneration of leg muscles. Muscle imaging has become an integrated part of the diagnostic workup of myopathies, and the recognized pattern of muscle involvement often points to the underlying gene defect [32]. The involvement of *sartorius* and *gracilis*, as opposed to most muscular dystrophies and the pronounced selective fatty infiltration in *semitendinosus*, *sartorius* and *gracilis* in the thighs and peroneal muscles at the calf level, are highly suggestive of a desminopathy, irrespective of the underlying mutation, disease stage or clinical muscle involvement [33].

The classical myopathologic findings of myofibrillar myopathies comprise myofibrillar disorganization beginning at the Z-discs, areas with reduction of oxidative enzyme activity (rubbed-out fibers and core-like lesions), abnormal accumulation of sarcoplasmic proteins that are stained with antibodies against desmin, aB-crystallin and myotilin, rimmed and non-rimmed vacuoles, and neurogenic-like abnormalities (small groups of atrophic fibers). At the ultrastructural level, there is myofibrillar disorganization and pathological protein aggregation [16]. In our patient’s muscle biopsy, we identified the typical signs of myofibrillar myopathy, herewith with prominent rubbed-out muscle fibers and dense granulofilamentous accumulations that are considered consistent with a desminopathy by electron microscopy [22,34]. 

The c.734A>G,p.(Glu245Gly) mutation in the *DES* gene found in our patient has already been reported in a large Indian kindred with 12 affected family members presenting with skeletal myopathy, severe heart disease and myofibrillar alterations in muscle biopsy. Nevertheless, in the absence of mRNA analysis, the authors could only speculate about the effect of the mutation at the protein level and they suggested that it either introduces the p.Glu245Gly missense mutation, or it results in a putative in-frame skipping of exon 3 corresponding to the deletion of residues p.Asp214-Glu245 [28]. By performing RNAseq on an RNA sample from the quadriceps muscle biopsy, we showed that this variant provokes a defect in exon 3 splicing and thus should be considered clearly pathogenic. Of note, the end of exon 3/beginning of intron 3 harbors pathogenic variants associated with myopathy and cardiomyopathy and all splice-site mutations of the *DES* gene that result in exon 3 skipping are located in this site (i.e., c.735G>T,p.(Glu245Asp), c.735G>C,p.(Glu245Asp), c.735+1G>A, c.735+1G>T, c.735+3A>G, c.735G>C). These mutations cause an in-frame skipping of exon 3, leading to the fusion of exons 2 and 4. [18,24,25,26,27,35,36,37]. Expression studies in SW13 (vim-) cells have showed that this mutant desmin product, lacking 32 amino-acids (from Asp214 through Glu245), is not functional and aggregates in desmin positive material in patients’ muscle [36]. It should be also noted that in this portion of the desmin protein resides the binding site for nebulin and, most probably, it is important for the linkage of the myofibrillar Z-discs to the intermediate filaments [38]. The clinical phenotypes in the reported patients consisted of cardiomyopathy (restrictive, dilative or rarely a phenotype transitioning from a hypertrophic to a restrictive and finally to a dilated type) preceding, as in the present case, the occurrence of skeletal myopathy [24,26,27,36]. 

In conclusion, the c.734A>G,p.(Glu245Gly) heterozygous variant in the *DES* gene is associated with myofibrillar myopathy and cardiomyopathy by exon 3 skipping.

## 4. Materials and Methods

### 4.1. Muscle Biopsy

Muscle specimen was obtained from the left quadriceps muscle by open biopsy, under local anesthesia. A portion of the muscle sample was snapped frozen in liquid-nitrogen cooled isopentane. Six-μm thickness cryostat sections were cut and used for histological, histochemical and immunohistochemical studies, using conventional techniques. Histological and histochemical reactions included hematoxylin–eosin (HE), Gomori’s modified trichrome, NADH-TR, SDH, cytochrome oxidase (COX), SDH-COX, ATPase (pH 9.4), phosphorylase, NSE, periodic acid Schiff (PAS), diastase-PAS, and Oil red O stains. Immunohistochemical study included the following primary mouse monoclonal antibodies: anti-desmin (clone D33, 1:70, Richard-Allan Scientific, Kalamazoo, MI, USA), anti-myotilin (clone RSO34, 1:20, LEICA Biosystems Newcastle Ltd., Newcastle upon Tyne, UK). Sections were incubated for 1 h at room temperature with the primary antibody and immunostained with biotin-extravidin (Extra-2, Sigma, St. Louis, MO, USA). Color was developed with amino-ethyl-carbazole (AEC). For conventional electron microscopy, fresh muscle samples were fixed in buffered 2.5% glutaraldehyde—2% paraformaldehyde, post-fixed in 1% osmium tetroxide and embedded in fresh epoxy resin mixture. Ultrathin sections at a thickness of 80 nm were stained with uranyl acetate and lead citrate and examined with a Philips 420 transmission electron microscope, and images were acquired with a Megaview G2 CCD camera (Olympus SIS, Münster, Germany).

### 4.2. Targeted Gene Enrichment, Next Generation Sequencing NGS (High-Throughput Sequencing)

Patients’ DNAs were extracted from peripheral blood with QIAsymphony (Qiagen, Hilden, Germany) and qualitatively checked using Tape Station DNA genomic array (Agilent, Santa Clara, CA, USA). Custom-targeted gene enrichment and DNA library preparation were performed using the NimbleGen EZ Choice probes and Kappa HTP Library preparation kit according to the manufacturer’s instructions (NimbleGen, Roche Diagnostics, Madison, WI, USA). A specific custom panel of 17 genes was designed including genes associated with myofibrillar myopathies. The RefSeq coding sequences were determined as consensual for genetic diagnosis within a French nationwide working group [39]. The targeted regions include all coding exons and ±50 base pairs of flanking intronic regions of 17 genes known to be involved in myofibrillar myopathies (Table 1). Paired-end sequencing was performed on a 250-cycle Flow Cell (Illumina, Santa Cruz, CA, USA) using the Illumina MiSeq platform. Eight libraries were multiplexed per run.

### 4.3. Bioinformatics Analysis

MiSeq Software (MiSeq Control Software v. 2.6.2.1) generates FASTQ format files after demultiplexing patients’ sequences. Sequence alignment against the human reference genome (Hg19) was performed using BWA-MEM. Variant calling was performed using the GATK Haplotype Caller program. Detected variants were then annotated using ANNOVAR and CADD tools. Detected variants with sequencing depth greater than 30× and with at least 20% of reads supporting the alternative allele were kept for analysis. Detection of copy number variation (CNV) was performed after coverage normalization, by computing the ratio of a target’s coverage of a given individual over the mean coverage of this target across all patients of the same sequencing run.

### 4.4. Variants Interpretations

Pathogenicity of variants was determined according to current ACMG guidelines [40]. Variants were filtered out according to their allele frequency (≤1%) as reported in the GnomAD database (http://gnomad.broadinstitute.org/, accessed on 1 April 2020) We then evaluated each variant considering a review of the literature, the location of the variant in the gene and the resulting corresponding protein, the in silico prediction tools (Polyphen2, SIFT, GVGD and CADD for missense variants and SpliceSiteFinder like, MaxEntScan, NNSPLICE, GeneSplicer and Human Splicing Finder for splicing variants) and functional studies when available. The SuSPect method (http://www.sbg.bio.ic.ac.uk/suspect, accessed on 1 April 2020) was also used for prediction. In addition, we looked at a local database of pathogenic variants related to our experience on the molecular diagnosis of myopathies. All variants considered as pathogenic and likely pathogenic have been confirmed by a second independent method (Sanger sequencing).

### 4.5. RNA Sequencing

Muscular quadriceps biopsy of the patient was realized. RNA extraction was done (Maxwell RSC simply RNA Blood Kit, Promega, WI, USA) and RNA integrity (RIN) was evaluated with the TapeStation 4200 device. If RNA achieved an RIN score >6, it was considered utilizable for RNAseq. RNA was diluted to obtain 25 µL to 12 ng/µL. Then, the Illumina Stranded mRNA Prep protocol (Illumina, San Diego, CA, USA), which converts mRNA in dual-indexed libraries, was followed. Briefly, Oligo(dT) magnetic beads purified and captured RNA with poly-A tails. Then, the purified mRNA was fragmented and a first-strand complementary DNA (cDNA) was synthetized. During second-strand cDNA synthesis dUTP replaced dTTP. The final step ligated adapters to fragment ends. The resulting products were purified and selectively amplified for the sequencing realized with a NextSeq 500 using 75 bp paired-end reads. Raw data from the Illumina NextSeq 500 sequencer (Illumina) was converted to FASTQ and demultiplexed with bcl2fastq. Reads from each sample were aligned to the human hg38 reference genome using the STAR tool. The quality metrics of the two previous software were supplemented by those of FastQC, Picard RnaSeqMetrics and RNA-SeQC and brought together in an HTML file thanks to MultiQC. The analysis of aberrant transcripts was carried out with visualization of Sashimi plots in IGV (Integrative Genomic Browser, version 2.11.1) on the regions of interest previously identified after DNA sequencing.

## Figures and Tables

**Figure 1 ijms-24-11181-f001:**
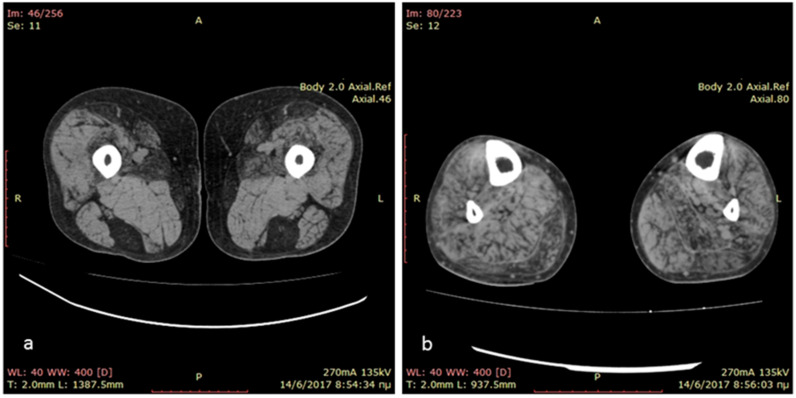
Muscle computed tomography of the thighs showing predominant involvement of *semitendinosus*, *gracilis*, *sartorius* and *adductor longus* muscles in the thighs (**a**) and diffuse fatty degeneration of leg muscles (**b**).

**Figure 2 ijms-24-11181-f002:**
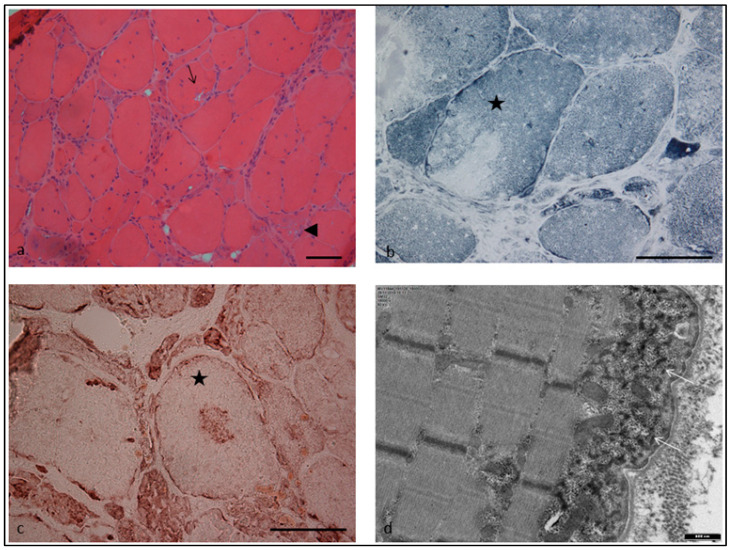
Skeletal muscle section from left quadriceps muscle biopsy. (**a**): H&E stain showing variable muscle fiber size with increased internal nuclei, rimmed vacuoles (arrow) and grouped atrophic fibers giving a pseudo-neurogenic aspect (arrowhead) (×20). (**b**): NADH-TR stain showing a rubbed-out area in the muscle fiber (star) (×40). (**c**): Immunohistochemistry with the anti-human desmin monoclonal mouse antibodies (sc-23879, Santa Cruz) showing a muscle fiber with desmin-positive sarcoplasmic aggregates (star) (×40) (**d**): Electron microscopy: Subsarcolemmal granulofilamentous material (arrows) (×18,000). Scale bar 50 μm in panels (**a**–**c**).

**Figure 3 ijms-24-11181-f003:**
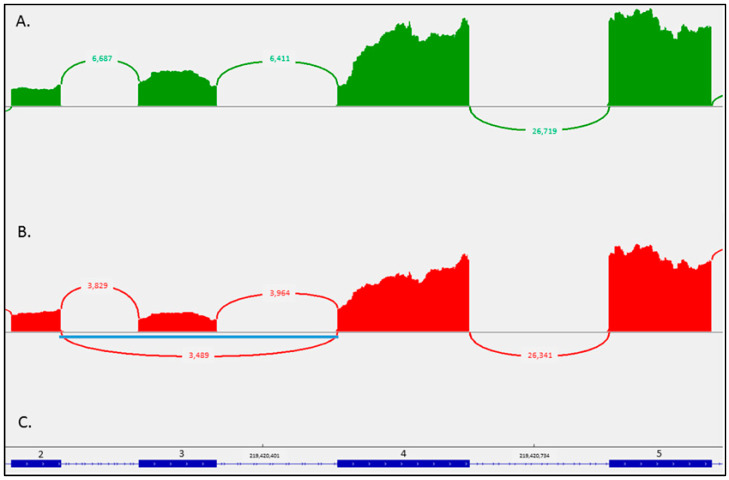
Exon 3 splicing defect of *DES* transcript (NM_001927.4) analyzed by RNAseq. (**A**). Sashimi-plot of mRNA sequencing showing exon 3 splicing in control muscle and (**B**). in the proband muscle. The IGV histograms represent read coverage of the exons and arcs indicate the number of junction-spanning reads supporting the exons junction. The aberrant junction resulting from the c.734A>G variant is indicated by a blue line indicating the heterozygous deletion of exon 3. (**C**). Exon-intron zoom of *DES* gene (blue solid rectangle for numerated exons 2 to 5 (among 9 exons) and line for intron).

**Table 1 ijms-24-11181-t001:** NGS panel of genes associated with myofibrillar myopathies.

*ACTA1* (NM_001100.3)
*BAG3* (NM_004281.3)
*CRYAB* (NM_001885.2)
*DES* (NM_001927.3)
*DNAJB6* (NM_058246.3)
*FHL1* (NM_001159702.2)
*FLNC* (NM_001458.4)
*GNE* (NM_001128227.3)
*HSPB1* (NM_001540.3)
*HSPB8* (NM_014365.2)
*MYH2* (NM_017534.5)
*MYOT* (NM_006790.2)
*RYR1* (NM_000540.2, with partially covered exon 91)
*SQSTM1* (NM_003900.4)
*TTN* (NM_001267550.1)
*VCP* (NM_007126.3)
*ZASP/LDB3* (NM_001080114.1, NM_007078.2, NM_001171610.1)

## Data Availability

The data presented in this study are available on request from the corresponding author.

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
