# Peer review of "Deep Characterization of a Greek Patient with Desmin-Related Myofibrillar Myopathy and Cardiomyopathy"

_ijms, 2023, doi:10.3390/ijms241311181_

Round 1
Reviewer 1 Report
In the case report 'Deep characterization of a Greek patient with a desmin-related myofibrillar myopathy and cardiomyopathy' submitted by Papadopoulos et al. to IJMS, the authors identified a single nucleotide polymorphism leading to a slice defect within the DES gene. The phenotype of the patient is a combination of a skeletal muscle myopathy and a restrictive / hypertrophic cardiomyopathy.
In general, this manuscript is interesting and deserves publication. However, some points can be improved by the authors and I suggest in summary a major revision.
1.) Abstract: Line 16: please add the word 'protein' after intermediate filament.
2.) Do not mix the intercalated disk with the Purkinje fibers. Intercalated disk are subcellular structures of the cardiomyocytes whereas Purkinje fibers are structures at the organ level. Please seperate these two entities.
3.) Please reference an example for recessive inheritance (line 40, introduction). For example the missense mutation Y122H was recently found in a similar patient with restrictive cardiomyopathy and is also leading to a filament assembly defect.
4.) Could you please add references for every of the following cardiomyopathies, which you mention in line 53 of the introduction: arrhythmogenic, hypertrophic, dilated and restrictive CM cardiomyopathies.
5.) Could you also add non-compaction cardiomyopathy in line 53 including a reference. Desmin-p.A337P (DES-p.A337P) was recently described in a patient with Left-ventricular non-compaction cardiomyopathy. This should be cited.
6.) Please increase the size of Figure 1. It is difficult to read.
7.) In Figure 2 the scale bars are missing. Please add them.
8.) Please write all human gene names in Italics in the complete manuscript including tables.
9.) Please show also the Sashimi-plot of the control, which you used (Figure 3).
10.) Recently, a different variant in close proximity (c.735G>C) was discovered and characterized in detail leading to a similar splice defect, which you discovered in your patient. The authors of this study used nanopore amplicon sequencing. The phenotype of this patient is pretty similar to the described one by your group (severe restrictive cardiomyopathy, including heart transplation). Could you compare and discuss this case with your patient. This would support the same conclusion of both manuscripts.
11.) The labeling of the sub-chapters (all are 1.1) is incorrect. Please correct them.
In summary, the authors present an interesting case with an interesting mutation in the DES gene leading to a splice defect, although it is localized at the end of exon-3 and not directly affecting the splice site nucleotides. The manuscript can be improved by adding and discussing some more literature about similar cases. In the introduction it would be maybe also helpful to cite the following review article dealing about desmin-caused cardiomyopathies: "Molecular insights into cardiomyopathies associated with desmin (DES) mutations." Biophysical reviews 10.4 (2018): 983-1006.
However, I am pretty optimistic that the authors can fix these points in a major revision. Good luck!
English should be read and corrected by a native speaking editor.
Author Response
Response to reviewer 1.
Dear reviewer 1,
We thank you for the thorough review of our manuscript and for your comments that we tried to address:
- Abstract: Line 16: please add the word 'protein' after intermediate filament.
Response: we added the word “protein”
- Do not mix the intercalated disk with the Purkinje fibers. Intercalated disk are subcellular structures of the cardiomyocytes whereas Purkinje fibers are structures at the organ level. Please separate these two entities.
Response: we rephrased appropriately
- Please reference an example for recessive inheritance (line 40, introduction). For example the missense mutation Y122H was recently found in a similar patient with restrictive cardiomyopathy and is also leading to a filament assembly defect.
Response: we added a reference for recessive inheritance (Ref 4)
- Could you please add references for every of the following cardiomyopathies, which you mention in line 53 of the introduction: arrhythmogenic, hypertrophic, dilated and restrictive CM cardiomyopathies.
Response: we added the appropriate reference for each type of cardiomyopathies (Ref 10-14)
- Could you also add non-compaction cardiomyopathy in line 53 including a reference. Desmin-p.A337P (DES-p.A337P) was recently described in a patient with Left-ventricular non-compaction cardiomyopathy. This should be cited.
Response: we added non compaction cardiomyopathy in line 53 and we cited the article (Ref 14)
- Please increase the size of Figure 1. It is difficult to read.
Response: we increased the size of Figure 1, as well as the size of the rest of the figures
- In Figure 2 the scale bars are missing. Please add them.
Response: we added the scale bars
- Please write all human gene names in Italics in the complete manuscript including tables
Response: we wrote all human gene names in Italics (including Tables)
- Please show also the Sashimi-plot of the control, which you used (Figure 3).
Response: we added the Sashimi-plot of the control in Figure 3 and we modified accordingly the text
- Recently, a different variant in close proximity (c.735G>C) was discovered and characterized in detail leading to a similar splice defect, which you discovered in your patient. The authors of this study used nanopore amplicon sequencing. The phenotype of this patient is pretty similar to the described one by your group (severe restrictive cardiomyopathy, including heart transplation). Could you compare and discuss this case with your patient. This would support the same conclusion of both manuscripts.
Response: we added this report, as well as the mutation in our manuscript (Ref 37)
- The labeling of the sub-chapters (all are 1.1) is incorrect. Please correct them.
Response: we corrected the labeling of the sub-chapters
In summary, the authors present an interesting case with an interesting mutation in the DES gene leading to a splice defect, although it is localized at the end of exon-3 and not directly affecting the splice site nucleotides. The manuscript can be improved by adding and discussing some more literature about similar cases. In the introduction it would be maybe also helpful to cite the following review article dealing about desmin-caused cardiomyopathies: "Molecular insights into cardiomyopathies associated with desmin (DES) mutations." Biophysical reviews 10.4 (2018): 983-1006.
Response: we cited the suggested review (Ref 15)
Reviewer 2 Report
In this case report, Papadopoulos, et al reported a patient with myofibrillar myopathy with a detailed clinical time-course over 30 years, showed several findings of clinical imaging. The authors performed genetic testing and confirmed that the identified heterozygous c.734A>G variant caused an in-frame defect in exon 3 via RNA-sequencing analysis using clinically obtained skeletal muscle sample. The epidemiological background of desminopathies, interpretation of the histopathological findings base on the structural basis of desmin are sufficiently discussed. Combined with the previous report showing the familial case carrying the same variant, the authors’ findings will be of great help for clinicians engaged in neuromuscular disorder and cardiovascular disease. I have some minor comments.
Please show the reference of the epidemiological data demonstrating that desminopathies affecting up to 5 individuals in 10,000 (line 45).
Immunohistochemical analysis using myotilin antibody was described both in results and methods sections, but the immunohistochemical result is not shown.
Are there any pathogenic or likely-pathogenic variants identified in the targeted sequencing of a multigene panel other than DES?
Author Response
Dear reviewer,
We thank you for the thorough review of our manuscript and for your comments that we tried to address:
In this case report, Papadopoulos, et al reported a patient with myofibrillar myopathy with a detailed clinical time-course over 30 years, showed several findings of clinical imaging. The authors performed genetic testing and confirmed that the identified heterozygous c.734A>G variant caused an in-frame defect in exon 3 via RNA-sequencing analysis using clinically obtained skeletal muscle sample. The epidemiological background of desminopathies, interpretation of the histopathological findings base on the structural basis of desmin are sufficiently discussed. Combined with the previous report showing the familial case carrying the same variant, the authors’ findings will be of great help for clinicians engaged in neuromuscular disorder and cardiovascular disease. I have some minor comments.
- Please show the reference of the epidemiological data demonstrating that desminopathies affecting up to 5 individuals in 10,000 (line 45)
Response: we rephrased appropriately. The point of this sentence was to mention that desminopathies are considered rare diseases and thus they affect no more than 5 individuals in 10,000
- Immunohistochemical analysis using myotilin antibody was described both in results and methods sections, but the immunohistochemical result is not shown
Response: Immunohistochemistry for a battery of antibodies against protein involved in myofibrillar myopathies revealed the presence of both desmin and myotilin sarcoplasmic immunoreactive aggregates. We decided to only show a Figure of desmin immunoreactive aggregates, as we believe that it is sufficient to illustrate the protein aggregate pathology. We added “not shown” after myotilin in the results section
Are there any pathogenic or likely-pathogenic variants identified in the targeted sequencing of a multigene panel other than DES?
Response: there were no other pathogenic or likely-pathogeni variants in any other gene included in the multigene panel
Round 2
Reviewer 1 Report
Congratulations. The authors have improved their manuscript and it deserves publication in IJMS.